# Health Status and Stress in Different Categories of Racing Pigeons

**DOI:** 10.3390/ani11092686

**Published:** 2021-09-13

**Authors:** Marjan Kastelic, Igor Pšeničnik, Gordana Gregurić Gračner, Nina Čebulj Kadunc, Renata Lindtner Knific, Brigita Slavec, Uroš Krapež, Aleksandra Vergles Rataj, Olga Zorman Rojs, Barbara Pulko, Maša Rajšp, Nina Mlakar Hrženjak, Alenka Dovč

**Affiliations:** 1BUBA d.o.o.-Veterinary Clinic and Pet Supply Store, Rožna dolina 5, 1290 Grosuplje, Slovenia; vet.kastelic@siol.net; 2Institute for Poultry, Birds, Small Mammals and Reptiles, Veterinary Faculty, University of Ljubljana, Gerbičeva 60, 1000 Ljubljana, Slovenia; igorpsenicnik88@gmail.com (I.P.); renata.lindtnerknific@vf.uni-lj.si (R.L.K.); brigita.slavec@vf.uni-lj.si (B.S.); uros.krapez@vf.uni-lj.si (U.K.); olga.zorman-rojs@vf.uni-lj.si (O.Z.R.); barbara.pulko@gmail.com (B.P.); masa.rajsp95@gmail.com (M.R.); nina.mlakar@vf.uni-lj.si (N.M.H.); 3Department of Animal Hygiene, Behavior and Animal Welfare, Faculty of Veterinary Medicine, University of Zagreb, Heinzelova 55, 10000 Zagreb, Croatia; ggracner@gmail.com; 4Institute of Preclinical Sciences, Veterinary Faculty, University of Ljubljana, Gerbičeva 60, 1000 Ljubljana, Slovenia; nina.cebulj.kadunc@vf.uni-lj.si; 5Institute of Microbiology and Parasitology, Veterinary Faculty, University of Ljubljana, Gerbičeva 60, 1000 Ljubljana, Slovenia; aleksandra.verglesrataj@vf.uni-lj.si

**Keywords:** *Columba livia domestica*, infectious diseases, serum corticosterone, welfare

## Abstract

**Simple Summary:**

Corticosterone is the most important “stress” hormone in birds. Stress response is influenced by different factors, such as phylogeny, feed supply, age, body condition, health status, climate, predators. Pigeon races over long distances, 500 km or more, can lead to the “exploitation” of animals if not strictly regulated and observed, jeopardizing their welfare status. Animals should be in good health and body condition, and health monitoring must be implemented. In stressful situations such as races, the possibility of infection increases. Clinically asymptomatic infections can flare up later in the breeding season and can cause high offspring mortality. For example, infections with circoviruses are particularly important because of their ability to weaken the immune system. The purpose of this work is to identify the critical stress points during the active training season of racing pigeons for the improvement of their condition and the preservation of their welfare during races. The aim of our study was to determine the serum corticosterone levels in different categories of racing pigeons exposed to severe stress factors. At the time of racing, some parameters of stress, including environmental factors, or the presence of infectious diseases or parasites, were recorded. It was found that participation in the race significantly increased serum corticosterone levels and remained high even one month after the race. Therefore, training and races should be properly managed and planned.

**Abstract:**

The influence of different stress parameters in racing pigeon flocks, such as the presence of diseases and environmental conditions at the time of the races, were described. A total of 96 racing pigeons from 4 pigeon flocks were examined, and health monitoring was carried out. No helminth eggs and coccidia were found. *Trichomonas* sp. was confirmed in subclinical form. Paramyxoviruses and avian influenza viruses were not confirmed, but circovirus infections were confirmed in all flocks. *Chlamydia psittaci* was confirmed in one flock. Blood samples were collected, and HI antibody titers against paramyxoviruses before and 25 days after vaccination were determined. To improve the conditions during racing and the welfare of the pigeons, critical points were studied with regard to stress factors during the active training season. Serum corticosterone levels were measured in the blood serum of four different categories of pigeons from each flock. Corticosterone levels were almost twice as high in pigeons from the category that were active throughout the racing season, including medium- and long-distance racing, compared to the other three categories that were not racing actively. Within five hours of the finish of a race, the average serum corticosterone level was 59.4 nmol/L in the most physically active category. The average serum corticosterone level in this category remained at 37.5 nmol/L one month after the last race.

## 1. Introduction

Pigeons are naturally gifted with the ability to find “home” from distant places, relying on abilities beyond memory. Humans discovered this capability by accident and then began to breed them selectively [1]. The breeding of domestic pigeons is one of the most rapidly developing areas of the animal world. The great races attract breeders from dozens of countries every year [1,2].

Shows and races with pigeons often lead to exploitation, injury, and death if not strictly regulated and observed [3]. Good body condition and stress resistance usually depend on feed supply, but health status throughout the year is also of great importance. Transport to the race and environmental factors during the race can be very stressful for pigeons, and sometimes a high number of pigeons perish. Warzecha [4] indicated that these problems can affect many animals, and government veterinarians should be actively involved in these activities.

In many species, including birds, rodents, reptiles, and amphibians, corticosterone (CORT) is the main glucocorticoid involved in the regulation of energy, immune, and stress responses. Responses to chronic stressor exposure and chronically elevated glucocorticoids include reduced growth, immunocompetence, reproduction, and survival. The effects of elevated glucocorticoids have an influence on survival, physiological, behavioral, reproductive, and intergenerational responses in wild vertebrates [5,6].

CORT is the major “stress” hormone in birds, with short-term changes mediating adaptive behavioral and metabolic responses to adverse environmental events (increased effort, transport, predators, fasting, and climatic conditions) as well as health status [7,8,9].

Because of diurnal rhythms of plasma corticosterone levels, the time of sampling is important [10,11]. However, in birds, the maximum level of CORT naturally occurs at daybreak [10]. Lumeij et al. [12] determined that baseline serum corticosterone concentrations in racing pigeons varied from less than 0.2 to 1.24 μg/dL (5.77–35.77 nmol/L) after 24 h of rest.

Various stressors can also occur during transport to and during the race, especially if it is not strictly regulated. Inappropriate environmental factors during the race, such as air temperature, relative humidity, air velocity, and magnetic radiation, can lead to additional stress [13,14]. 

Stressful situations during races also increase the possibility of infections. Clinically asymptomatic infections can flare up later in the breeding season and cause great losses of offspring [15]. The major bacterial pathogens in a racing flock are *Salmonella typhimurium* var. Copenhagen, *Escherichia coli*, and a group of bacteria that cause chronic respiratory disease and lead to poor performance, mostly caused by *Chlamydia psittaci* (CP), *Pasteurella*, and *Mycoplasma* species. In addition, some fungi and yeasts (e.g., *Aspergillus*, *Candida*, *Cryptococcus*), endoparasites (e.g., *Eimeria*, *Haemoproteus*, *Trichomonas*), and ectoparasites (e.g., Mallophaga, Hippoboscid pigeon flies) molested birds, stressing them and causing various diseases [16,17]. Zigo [2] found an increased incidence of endoparasite infestation and respiratory syndrome at the time of racing.

Diseases caused by viruses, whether clinical or subclinical, play an important role in the occurrence of stress. The most often detected groups of viruses include paramyxoviruses (avian paramyxovirus 1) (APMV-1), circoviruses (pigeon circoviruses) (PiCV), adenoviruses, herpesviruses, and poxviruses [18]. Avian influenza viruses (AIV) are less often detected in pigeons. They play a minor role in the epidemiology of H5 influenza. In pigeons, influenza A virus of subtype H7 can cause conjunctivitis, tremor, paresis of wings and legs, and wet droppings. Nevertheless, free-flying domestic pigeons can act as mechanical vectors and vehicles for long-distance transmission of any influenza A virus, if plumage or feet are contaminated [19,20].

Among viruses, PiCV is the most frequently detected among pigeons [21], which is important due to its ability to weaken the immune system. The main consequences of PiCV infection are atrophy and other pathophysiological changes to organs of the immune system (e.g., bursa of Fabricius, thymus, spleen gut-associated lymphoid tissue, bronchus-associated lymphoid tissue, bone marrow, liver, kidney, larynx, trachea, lung, small and large intestine, pancreas). It has been established that infection with pigeon circovirus leads to apoptosis of lymphocytes. For the reasons mentioned above, PiCV is considered as a potential immunosuppressive agent [22]. PiCV infections are capable of predisposing birds to concomitant infections with other pathogens [23].

The aim of this study was to assess CORT levels in different categories of racing pigeons exposed to severe stress factors in order to determine critical stress points during the active season, improve conditions during racing flights and improve pigeon welfare by tracking certain stress parameters, such as the presence of infectious diseases or parasites, and determining environmental factors during the racing season.

## 2. Materials and Methods

Four flocks of racing pigeons from different breeders were included in the study. Each pigeon flock consisted of 100 to 150 parent racing pigeons. All flocks had similar husbandry conditions and were fed with diets produced by the same manufacturer. Active pigeons, which participated in training and races, always flight together. Transport to the trainings and games was also similar.

### 2.1. Pigeons; Sampling

A total of 96 racing pigeons from 4 breeders (24 per breeder) were examined for various stress parameters during the racing season. Four different categories (6 pigeons from each group) were compared in each pigeon flock. The first category/group (G1) consisted of sexually mature breeding pigeons not included in training or races. The second group (G2) contained young pigeons (less than 1-year-old) that did not participate in training. The third group (G3) contained pigeons that participated in training but not in medium- or long-distance flights. In the fourth group (G4), racing pigeons used in training and on the medium- and long-distance flights were included. These birds were active throughout the racing season. The pigeons from G3 and G4 were 2 to 7 years old. All the pigeons had identification rings.

A total of 24 pigeons from G4 group (6 from each flock) were tested for CORT levels twice: within 3 to 5 h after returning from the last race (G4a) and 30 days after the last race in the season (G4b). Pigeons from all categories were also tested for CORT concentration in both samplings.

To cheque the health status before the last race, the pigeons were clinically examined, and various samples were taken for further laboratory analyses. Common fecal samples were collected for each group separately (N = 16, 4 from each flock, 1 for each group) for intestinal parasites testing. Samples were transported to the laboratory in transport bags at 4 °C. Additionally, 6 oropharyngeal samples were collected for *Trichomonas* sp. testing per group. Transport of samples to the laboratory was carried out in transport bags at room temperature, and samples were examined within 6 h.

Cloacal and oropharyngeal samples (Copan swabs, Brescia, Italy) and blood samples (blood collection tubes, BD Microtainer^®^, SST™, Monroe, LA, USA) were collected from each bird individually before (G1, G2, G3) the last race and immediately after the race was over (G4). In all categories/groups (G1, G2, G3, and G4), 6 cloacal and 6 oropharyngeal samples were collected for PiCV, APMV, AIV, and CP determination. Additionally, 6 oropharyngeal samples (for each group) for *Trichomonas* sp. and 6 pools of feces (for each group) for endoparasites were collected. Blood samples in a volume of 0.5 mL were obtained by venipuncture of the ulnar cutaneous vein and collected in microtainer tubes using a serum separator (Becton Dickinson, Heidelberg, Germany). Swabs, feces, and blood samples were transported to the laboratory at 4 °C. Swabs were stored at −20 °C. A hemagglutination inhibition assay (IHA) was performed within 48 h after collection. Serum was used to determine HI antibody titers against paramyxoviruses and the immunity status of the presumably vaccinated pigeons. The first sampling results did not show satisfactory protection. Thus, all 4 flocks were revaccinated 5 days after the last race was finished. The Chevivac-P200 vaccine (Chevita GmbH, Pfaffenhofen, Germany) was used for vaccination. The vaccine was administered strictly subcutaneously dorsally in the neck toward the tail but not immediately behind the head, according to the manufacturer’s instructions. The effect and responsiveness of vaccination were assessed after 25 days (i.e., 30 days after the last race) and is reported in the text as group G4b.

### 2.2. Laboratory Tests

Cloacal and oropharyngeal swabs were used for molecular detection of pathogenic viruses and bacteria. Swabs were vortexed individually in 2 mL of PBS for 2 min, and 100 µL aliquots of each swab were pooled to produce 300 µL samples for DNA and RNA extraction. Pools were prepared in sterile PBS from 3 samples of cloacal or oropharyngeal swabs collected from each pigeon group (G1, G2, G3, and G4; 2 pools for each group) and for each flock separately (N = 64: 32 cloacal and 32 oropharyngeal pools). Total DNA and RNA were extracted from 140 µL of the pooled samples using the QIAamp Viral RNA Mini Kit (Qiagen, Hilden, Germany) according to the manufacturer’s instructions. Previously published molecular methods were used to detect pathogens in the samples collected in the study: PiCV [24], APMV-1 [25], AIV [26] and CP [27]. A species-specific real-time PCR assay was used for further determination of CP [28].

Pooled fecal samples (for each group) were examined for the presence of endoparasites using the flotation and sedimentation method [29].

*Trichomonas* sp. was detected microscopically in freshly prepared wet mounts. If no trichomonas was present in the observed sample, a drop of iodine solution was added, and the sample was re-examined.

IHA for HI antibody titers against APMV-1 was performed as described in a previous study [30].

CORT was measured in serum using a commercial enzyme immunoassay Corticosterone ELISA (Demeditec Diagnostics GmbH, Kiel, Germany) according to the manufacturer’s instructions.

### 2.3. Last Race

The last race was from Timisoara, Romania, to Ljubljana and its surroundings, Slovenia, 155 pigeons participated. The average flight distance for pigeons from G4 was 504.91 km. The pigeons were released at 12:20 on 9 July 2017 (18°45′43″ E, 45°49′03″ N), and the race was completed at 6:41 on 10 July 2017. Speed results were calculated based on the speed of the first 5 pigeons. According to the data provided by breeders through internet applications [31] the weather was clear, and the air temperature was very high (34 °C at 2 m above the ground).

During the last race, an animal welfare expert followed the procedures during transport, and he was present with the driver.

### 2.4. Data Analysis

After data collection, mean values of serum corticosterone levels in different categories of racing pigeons were calculated. Next, non-parametric tests were applied to find possible statistically significant differences between individual groups (Mann–Whitney test, *p* < 0.05), and in the case of G4a and G4b groups, the differences immediately and 30 days after the race for the same group of pigeons (Wilcoxon test, *p* < 0.05). For those analyses, the Statistical Package for the Social Sciences, version 25 was utilized.

## 3. Results

### 3.1. Health Status of the Pigeon Flocks

All the flocks were examined before the last race started. No helminth eggs were found in the fecal samples. Examination of the beak cavity confirmed the presence of *Trichomonas* sp. in all four flocks. However, none of them was found to have diarrhea or debris in the beak cavity. In all categories, the pigeons were clinically healthy and showed no presence of ectoparasites.

The presence of viruses (PiCV, APMV-1, AIV) and *Chlamydia psittaci* was checked by molecular methods in cloacal and oropharyngeal samples.

PiCV was detected in cloacal and oropharyngeal samples in all breeders but not in all categories. Among the adult breeding group (G1) and racing pigeons included in training and in races (G4), the virus was detected in two flocks. Among young pigeons (from G2 and G3), the virus was detected in three flocks. The results for PiCV are shown in Table 1.

APMV-1 and AIV were not detected. CP was confirmed in one cloacal sample in flock 2.

To determine paramyxovirus immune status in the flocks, sera samples were examined by the IHA for HI-antibody titers against APMV-1 (Table 2).

### 3.2. Serum Corticosterone Levels in Different Categories of Racing Pigeons

Within five hours after the race was finished, the average CORT level in G4a was 59.4 nmol/L, almost two-fold higher compared to those in the other three categories (G1, G2, and G3). In the same category, one month after the last race (G4b), CORT levels remained higher in two flocks compared to levels before the final race (Table 3 and Table 4).

Pairwise comparisons showed statistically significant differences for the following groups: G1 vs. G3, G2 vs. G3, G1 vs. G4a, G2 vs. G4a, and G3 vs. G4a (Table 5). Although there was no statistically significant difference found between G4a vs. G4b in the total sample, some differences emerged after we made comparisons for the individual flock separately (Table 5). Namely, significant differences in the measurements emerged for flocks 2 and 4 (in both cases Z = −2.201, *p* = 0.028).

### 3.3. Results Achieved in the Last Race

Transport from Ljubljana to Timisoara was controlled, and welfare was provided by using specially adapted trailers for pigeons. Climatic conditions were appropriate, and pigeons were observed every six hours, and the water supply was checked.

The average speed of the first five pigeons was 1037.94 m/min in flock 1, 1042.43 m/min in flock 2, 855.13 m/min in flock 3, and 1158.74 m/min in flock 4.

## 4. Discussion

### 4.1. Health Status of Pigeon Flocks

Pigeons are exposed to various stressors during the racing season. Increased stress during transport to the race and the race itself is an important factor that significantly affects their health status [2]. In stressful situations, the possibility of infection increases [13]. Clinically asymptomatic infections may flare up later in the breeding season and lead to major losses in the flock. Therefore, regular monitoring of flock health status is very important and should be conducted before and after the active season. In our study, antibodies against APMV-1 were assessed, and the presence of certain pathogens (PiCV, APMV-1, AIV, CP) that could cause disease or act as a stress factor in its subclinical form was investigated. It is also important to know the epidemiological situation of these pathogens in rural areas and surrounding countries. Pathogens, such as PiCV [32], APMV-1 [33,34], adenoviruses [34,35], AIV [34], CP [31,34], and *Trichomonas* sp. [36], were confirmed in different categories of pigeons in Slovenia. Transmission of the above-mentioned pathogens from feral pigeons to racing pigeons has been frequently noted in clinical practice.

The specificity of pigeon training and racing significantly impedes the principles of biosecurity [37]. Breeders in Slovenia train and race together, and, therefore, the flocks have closer contact, which could lead to transmission of viral pathogens (e.g., PiCV, APMV-1), bacterial pathogens (e.g., *Chlamydia psittaci*), or parasites (e.g., *Trichomonas* sp.). Another critical biosecurity issue is contact between racing pigeons and feral pigeons. Contact commonly occurs during long race flights, when pigeons have to rest, drink, and feed; these situations could result in the transmission of infections from feral pigeons.

In a previous study, 74.3% of cloacal and 54.1% of oropharyngeal samples collected from feral pigeons in Slovenia were positive for PiCV [32]. In another study of Slovenian racing pigeons, results showed that 93.3% of cloacal and 96.7% of oropharyngeal samples were positive for PiCV [34] and are comparable with results obtained in other countries [21]. The findings showed a high prevalence of infections with PiCV, a pathogen that could cause immunosuppression, which may favor secondary infections [38]. In the present study, PiCV was detected in all flocks but not in all categories in each flock. In the adult breeding group (G1) and pigeons included in training and racing (G4), the virus was detected in two flocks and in young pigeons (G2 and young pigeons from G3) in three flocks. Based on our limited results, we can only speculate that there is no direct correlation between PiCV infection and CORT levels.

The results for AIV were negative in our previous study and this study, which coincides with the results of other authors [19,39,40].

Chlamydial infections in feral pigeons in Europe and the focus on public health implications are described by Magnino et al. [41]. They found 19.4% to 95.6% seropositive pigeons and 3.4–50% PCR positive pigeons, indicating high importance of transmission to racing pigeons and also to humans. In our previous study of chlamydial infections in racing pigeons, CP was confirmed in cloacal swabs in 16.7% of samples. The determination of CP also indicates a high risk of infection to humans [39]. In this study, CP was confirmed in only one cloacal sample (2.1%). All oropharyngeal samples were negative.

Endoparasites could cause subclinical or even clinical disease. To prevent serious disease and consequently stress, pigeons must be treated regularly [9,16]. Zigo [2] found an increased incidence of coccidiosis (40.4%), trichomoniasis (17.3%), and other endoparasitoses (11.5%) at the time of racing. The results of parasitological examinations, performed just before the last race, showed that all samples were negative for helminths and coccidia, though *Trichomonas* sp. was confirmed in all flocks.

Due to the fact that APMV-1 is a serious viral pathogen that is endemic and is common in feral pigeons [17,37], breeders should regularly vaccinate pigeons to prevent infection and transmission. Vaccination is usually performed before and after the active flying season. Inactivated vaccines are used for the prophylaxis of APMV-1 in pigeons. Vaccines are based on different strains of paramyxoviruses, with the LaSota strain being one of the most commonly used [37]. Inactivated vaccines are administered by subcutaneous injection in various prevention programs. After the vaccination of 4- to 6-week-old seronegative pigeons with the inactivated LaSota vaccine in aqueous suspension, mean antibody titers were between 3 (log_2_) and 5 (log_2_) after 1 to 2 months. Titers between 2 (log_2_) and 3 (log_2_) were detected for 6 months, and thereafter, a decrease in titers was observed [42].

In our study, APMV-1 was not detected, but we did find that pigeons in all flocks, especially young pigeons less than one-year-old (G2), were rather poorly protected against this virus based on antibody titers detected by the IHA test (Table 1). The first IHA test found a titer of 3.5 (log_2_) in one flock, below 2.0 in two flocks, and as low as 0.0 in another flock. The flocks were vaccinated again 5 days after the last race, but only the active fliers (G4) were checked 25 days after immunization (i.e., 30 days after the last race). Titers were slightly elevated in flocks 1, 2, and 4 but still remained below 2.0 in flock 2 and 4. In flock 3, the titer remained 0.0. In this flock, the CORT level remained high after 30 days (65.4 nmol/L). A high CORT level (57.1 nmol/L) after 30 days was also found in flock 1, but the titer, in this case, increased from 2.3 to 5.3, and the minimum titer was 4. We can assume that there was no direct relationship between the vaccination response and stress. The reason for the low titer increase was probably due to the short testing interval after revaccination, which should be 1 to 2 months [42], or due to some other unknown stressors.

### 4.2. Serum Corticosterone Levels in Different Categories of Racing Pigeons

In all bird species studied, CORT is considered the most important glucocorticoid [12]. CORT measurements have been proven to be useful in measuring the welfare status of racing pigeons [43]. Romero and Wingfield [44] determined 2–9 ng/mL (5.77–25.96 nmol/L) for baseline serum CORT levels and 14–15 ng/mL (40.38–43.26 nmol/L) for stress-induced levels in free-living pigeons. Lumeij et al. [12] measured baseline concentrations of CORT in the serum of 30 racing pigeons after they had been kept quiet for 24 h. The average level was 0.34 μg/dL (9.81 nmol/L), and maximum level was 1.24 μg/dL (35.77 nmol/L).

Our results of CORT in serum were within the range of results obtained by other authors [12,44]. In three out of four flocks, the lowest level of CORT was found in the serum of pigeons that were only trained (G3). The average CORT level was 16.9 nmol/L (range from 9.1 to 25.4 nmol/L). In the fourth flock, the lowest level of CORT was found in the adult breeding group (G1) (13.2 nmol/L) and slightly higher in pigeons that were only trained (G3) (14.5 nmol/L). Active racing pigeons (G4) were the most stressed group. The average measured CORT level was 59.4 nmol/L (range from 37.3 to 69.0 nmol/L), and it seems that exhaustion during the race significantly increased the CORT level in serum.

Comparing active racing pigeons (G4) one month after the end of the last race, we found that in two flocks (flock 2 and flock 4), the levels of CORT dropped to levels lower than the levels of the whole flock during the active season, and in two flocks (flock 1 and flock 3), the levels were still higher than the levels before the last race. The levels remained high at 57.1 nmol/L in flock 1 and 56.4 nmol/L in flock 2. The reason could be further stress in the flock during this period or the values remaining high all the time. However, measurements need to be recorded more often to obtain the right answer.

The average values obtained from CORT in serum were the highest in flock 3 (44.02 nmol/L) and the lowest in flock 4 (18.31 nmol/L). When comparing the speed results of the last race, the pigeons from flock 3 had the worst race results, and the best results were obtained in flock 4. The average speed of the first five pigeons from flock 4 was 1037.94 m/min and 855.13 m/min from flock 3. We know that flock 4 (G3 and G4) had the most intensive training during the whole active season, which reflected in slightly higher CORT values compared to those in G1 and G2. The CORT level in the flock dropped below the pre-race level one month after the race. The breeder maintained a high level of health care after the race and used pills, electrolytes, and herbal teas at the time of the active season. This was also described in the literature [1] as a method to maximize the performance of pigeons and protect breeders’ investment.

### 4.3. Stress Factors during Transport to the Race and during the Race

Fast and appropriate transport to the race is very important. The number of pigeons in a box, supply of food and water during transport and at the launching place, as well as the supply of fresh air, should be strictly controlled and implemented [14]. The transport our pigeons received reached a high level of maximum support in terms of pigeon welfare. The only negative potential is that the common transport of several different flocks could lead to a possible transmission of diseases.

In addition, the duration and direction of the race should be included in the planning of each race, and environmental factors should be followed to ensure that the race occurs under suitable conditions that do not impact the welfare of pigeons. For these purposes, our breeders used the internet application called Ventusky by Mojzik and Prantl [24]. The chosen location for the last race was Timisoara, Romania, with an average flight distance of 504.91 km. During the race, the weather was clear, but the air temperature was very high, 34 °C at 2 m above the ground. Most, but not all of the pigeons that participated in the race returned to the pigeon houses within six days. Consequently, pigeon breeders prepared an internal protocol that allows racing only at temperatures up to 30 °C. The protocol was also adopted at the conclusion of a joint meeting of the Slovenian Pigeon Federation. The intention of this decisions is to improve the welfare of racing pigeons.

## 5. Conclusions

Race flights commonly result in elevated stress as measured by CORT levels and influence the welfare of pigeons that participate in or train for such events. Therefore, races and training should be properly managed and planned with pigeons thoroughly prepared for such challenges.

Only pigeons in good condition and those that are clinically healthy should be allowed to participate in the race. The presence of infectious diseases or parasites should be assessed before the start of each racing season. At the end of the racing season, pigeons may have elevated levels of stress hormones, such as CORT.

The distance, duration, and direction of the race should be planned according to environmental factors in order to reduce stress. It is necessary to avoid temperatures above 30 °C and to predict adverse weather conditions (storms, strong winds), pigeon exposure to predators, and unfavorable magnetic waves.

As training and racing can be very stressful for animals, strictly regulating the factors and circumstances that could jeopardize of racing animal welfare of racing pigeons should be a priority for those involved in the above-mentioned activities.

## Figures and Tables

**Table 1 animals-11-02686-t001:** The presence of PiCV in individual flocks and categories.

Flock	Category Inside the Flock ^1^	Circovirus Cloacal Sample	Circovirus Oropharyngeal Sample
Flock 1	G1	negative	negative
	G2	negative	negative
	G3	positive	positive
	G4	negative	negative
Flock 2	G1	positive	negative
	G2	positive	positive
	G3	negative	positive
	G4	positive	positive
Flock 3	G1	positive	positive
	G2	positive	positive
	G3	positive	positive
	G4	negative	negative
Flock 4	G1	negative	negative
	G2	positive	positive
	G3	negative	negative
	G4	positive	negative

^1^ G1: sexually mature breeding pigeons; G2: young pigeons, less than one year old, not participating in training; G3 pigeons participating in training but not in medium- or long-distance flights; G4: racing pigeons participating in training on medium- and long-distance flights.

**Table 2 animals-11-02686-t002:** HI-antibody titers (log^2^) against APMV-1.

Flock	Category Inside the Flock ^1^	Average of Antibody Titer (Range)
Flock 1	G1	1.7 (0–4)
	G2	0.0 (0)
	G3	0.0 (0)
	G4a	2.3 (0–4)
	G4b	5.3 (4–7)
Flock 2	G1	4.0 (1–7)
	G2	0.0 (0)
	G3	2.3 (0–7)
	G4a	0.7 (0–4)
	G4b	1.2 (0–4)
Flock 3	G1	1.8 (0–6)
	G2	0.0 (0)
	G3	0.7 (0–4)
	G4a	0.0 (0)
	G4b	0.0 (0)
Flock 4	G1	2.7 (0–5)
	G2	0.0 (0)
	G3	1.3 (0–2)
	G4a	0.8 (0–4)
	G4b	1.2 (0–3)
All flocks together	G1	3.5 (0–7)
	G2	0.0 (0)
	G3	1.7 (0–7)
	G4a	1.1 (0–4)
	G4b	2.2 (0–7)

^1^ G1: sexually mature breeding pigeons; G2: young pigeons, less than one-year-old, not participating in training; G3 pigeons participating in training but not in medium- or long-distance flights; G4a: racing pigeons participating in training on medium- and long-distance flights; G4b: racing pigeons participating in training on medium- and long-distance flights 25 days after immunization (i.e., 30 days after the last race). Pigeons from G4a and G4b were the same pigeons.

**Table 3 animals-11-02686-t003:** Serum CORT levels in different groups of pigeons in all flocks together.

Category/Group ^1^	Corticosterone Level nmol/L (SE) ^2^
G1	33.8 (6.04)
G2	28.3 (3.92)
G3	16.9 (3.78)
G4a	59.4 (10.62)
G4b	37.5 (7.89)

^1^ G1: sexually mature breeding pigeons; G2: young pigeons, less than one-year-old, they do not participate in training; G3 pigeons participated in training but not in medium- or long-distance flights; G4a: racing pigeons participated in training on the medium- and long-distance flights—tested within three to five hours after returning from the last race; G4b: racing pigeons participated in training on the medium- and long-distance flights tested 25 days after immunization (i.e., 30 days after the last race). Pigeons from G4a and G4b were the same pigeons. ^2^ SE: Standard error of the mean.

**Table 4 animals-11-02686-t004:** Serum CORT levels in different groups of pigeons are presented separately by individual flocks.

Flock (CORT Average nmol/L)	Category Inside the Flock ^1^	Corticosterone Level nmol/L (SE) ^3^
Flock 1 (40.50/6.72) ^2^	G1	36.5 (11.31)
	G2	34.3 (4.86)
	G3	9.1 (1.50)
	G4a	65.5 (18.34)
	G4b	57.1 (20.73)
Flock 2 (37.92/5.19) ^2^	G1	42.3 (7.67)
	G2	31.8 (8.10)
	G3	25.4 (14.04)
	G4a	69.0 (10.77)
	G4b	21.1 (7.15)
Flock 3 (44.02/8.85) ^2^	G1	43.1 (18.78)
	G2	32.6 (10.90)
	G3	13.0 (3.99)
	G4a	66.0 (34.53)
	G4b	65.4 (13.53)
Flock 4 (18.31/3.99) ^2^	G1	13.2 (3.79)
	G2	14.5 (4.84)
	G3	20.1 (3.94)
	G4a	37.3 (17.23)
	G4b	6.5 (1.52)

^1^ G1: sexually mature breeding pigeons; G2: young pigeons, less than one-year-old, they do not participate in training; G3 pigeons participated in training but not in medium- or long-distance flights; G4a: racing pigeons participated in training on the medium- and long-distance flights—tested within three to five hours after returning from the last race; G4b: racing pigeons participated in training on the medium- and long-distance flights tested 25 days after immunization (i.e., 30 days after the last race was finished). Pigeons from G4a and G4b were the same pigeons; ^2^ In the average estimation of the CORT levels immediately after the last race, only G4b was not included. ^3^ SE: Standard error of the mean.

**Table 5 animals-11-02686-t005:** Pairwise comparisons of categories (groups).

	Mann-Whitney Test									Wilcoxon Test
*Sig.*	G1 vs. G2	G1 vs. G3	G2 vs. G3	G1 vs. G4a	G2 vs. G4a	G3 vs. G4a	G1 vs. G4b	G2 vs. G4b	G3 vs. G4b	G4a vs. G4b
*Z*	−0.196	−2.753	−2.784	−2.052	−2.567	−4.228	−0.454	−0.227	−1.588	−1.867
*p*	0.845	0.006	0.005	0.040	0.010	<0.001	0.650	0.821	0.112	0.062

## Data Availability

Not applicable.

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
