# Peer review of "Health Status and Stress in Different Categories of Racing Pigeons"

_animals, 2021, doi:10.3390/ani11092686_

Round 1

Reviewer 1 Report

Dear Authors,

Thank you for submitting this paper on racing pigeons for review. I found the manuscript to be relatively well written and to cover to interesting topics relating to pigeon racing. From looking at the manuscript, I noticed that some aspects (parasitology and health) are not covered in the title. These could be included in the work. 

There is some merit to this work and some practical implication for those keeping pigeons. However, at current there are some revisions required to the work. I have included specific comments on the PDF version of the manuscript. In addition, please consider the following areas:

  1. Methods. Some aspects of the methods need to be covered in more detail, or are explained in other areas (e.g. results, discussion). Bring in more detail on aspects of the race and pigeon husbandry in the methods to ensure that the work is repeatable.
  2. Other influences on corticosterone. We know from studies on other animals that glucocorticoids can be influenced by variables other than stress. For example, breeding season and exercise can both increase glucocorticoid. While it is very likely the elevated CORT is a product of the racing, I would like to see more consideration of confounding factors.
  3. There are a few sentences structure / referencing errors. While I have corrected these where possible, a proof read would be valuable.

Author Response

DEAR REVIEWER 1

Thank you very much for your useful comments, suggestions and accurate changes. Corrections are made according to your suggestions as they follow.

We considered all your suggestions and changed (specific comments) in the article. We sent it back in pfd.file with short explanations.

General comments:

Answers to your additional comments.

  1. Some aspects of the methods need to be covered in more detail, or are explained in other areas (e.g. results, discussion). Bring in more detail on aspects of the race and pigeon husbandry in the methods to ensure that the work is repeatable.

We added some details in the text and also considered suggestions and comments made by reviewer 2.

  1. Other influences on corticosterone. We know from studies on other animals that glucocorticoids can be influenced by variables other than stress. For example, breeding season and exercise can both increase glucocorticoid. While it is very likely the elevated CORT is a product of the racing, I would like to see more consideration of confounding factors.

We added some data and new references (Sopinka et al, 2015; Romero et. al. 2009).

  1. There are a few sentences structure / referencing errors. While I have corrected these where possible, a proof read would be valuable.

References were checked and corrected.

Comment: Table 2 was changed. The contents of columns 3, 4 and 5 are grouped in column 3. Some data were added in tables.

Thank you again for your suggestions.

Sincerely yours author and coauthors.

Alenka Dovč

Reviewer 2 Report

Manuscript animals-1348059, entitled “Serum corticosterone levels in different categories of racing pigeons at the time of racing”

Recommendation:       The above paper is not suitable for publication in its present form.

General Comments:

  • This article provides useful information about the serum corticosterone levels in different categories of racing pigeons at the time of racing. However, there are a lot of grammar, stylistic and syntax errors. In some cases, these errors negatively influence the understanding of the text.
  • Please provide results and conclusion at the end of “Simple summary”
  • How were the differences in the examined parameters detected? Please provide SEM (means) or Q1-Q3 (medians) and P-values in Tables.
  • Several parts are repeated. Please delete them (L167-172, 275-276, 282-283, 428-430 etc)
  • L195-198: This part refers to Methods. Please delete or shorten it.
  • L257-263: This section should be removed in Materials and Methods in L157 (last paragraph)
  • L265-294: Please remove to the introduction (no results are presented)
  • L431-458: This part is not related with your results and is partly mentioned in introduction. Please shorten it and try to connect it with your results.
  • Authors often state that “Pigeons that are under stress are more susceptible to infections”. Did you reach to this conclusion in your study? Where are the data that confirm this conclusion?

Specific Comments:

L25, 69: “feed” instead of “food”

L27-28: “…welfare status. Animals should be in…”

L29: “such as” instead of “during”

L33: “…pigeons for the improvement of their condition and the preservation of their welfare…”

L36: “recorded” instead of “followed”

L57: Please rephrase “spheres of the animal world”?

L63-65: “Warzecha [4] indicated that these problems can affect many animals and government veterinarians should be actively involved in these activities.”

L67: “…(CORT) is the main…”

L67-68: “…immune and stress responses.”

L70-71: Please rephrase “and is associated with further support for adaptation of the stress response”

L72: “…the maximum level of CORT naturally occurs at daybreak…”

L75-76: “…especially if it is not strictly regulated [11,12].”

L87: “Zigo  [2] found…”

L99: “due to” instead of “given”

L109: “…improve conditions during racing flights and pigeon…”

L114: “per breeder” instead of “from each”

L121: Sexually mature?

L123-126: “Twenty-four pigeons from G4 group (six from each flock) were tested for CORT levels twice: within three to five hours after returning from the last race (G4a) and 30 days after the last race in the season (G4b). Pigeons from all categories were also tested for CORT concentration in both samplings.”

L128: Please delete “for”

L139: Please delete “Each of ninety-six pigeons was included in our study.”

L141: “…collected for PiCV, APMV, AIV and CP determination. Blood…”

L156: “provided the” instead of “were given”

L227-230: “Within five hours after the race was finished, the average CORT level in G4a was 59.4 nmol/L, almost two-fold higher compared to those in the other three categories (G1, G2 and G3).”

Table 3: Please provide subscripts “a” or “b” after “G4”

L256: This Table refers to CORT levels? What about the other parameters?

L265: “firstly” instead of “first”

L273-275: Please remove “[31]” after “Kerpal et al.” and delete “(2019)”

L278: “… have also been advanced [1].”

L290: “All these stressors…”

L294: “examined parameters” instead of “focus”

L298: “…health status and…”

L317: “collected” instead of “taken”

L328-331: Please rephrase

L343: “Due to the fact that” instead of “Because”

L369: Please delete

L375: “…welfare status of racing…”

L380: “…was 0.34 μg/dl (9.81 nmol/L) and the maximum level was…”

L383, 386: “trained” instead of “training”

L388: “Exhaustion” instead of “Extraction”

L396: “recorded” instead of “taken”

L423: “…protect breeders’ investment.”

L432: “The fast and appropriate transport to the race is very important."

Author Response

DEAR REVIEWER 2

Thank you very much for your useful comments, suggestions and accurate changes. Corrections are made according to your suggestions as they follow.

General Comments:

We considered your suggestions and changed all in the article. All minor corrections were done and paragraphs were rephrased according to your suggestions.

  1. This article provides useful information about the serum corticosterone levels in different categories of racing pigeons at the time of racing. However, there are a lot of grammar, stylistic and syntax errors. In some cases, these errors negatively influence the understanding of the text.

We have considered this statement.

  1. Please provide results and conclusion at the end of “Simple summary”

We added some results and conclusions in the simple summary.

  1. How were the differences in the examined parameters detected? Please provide SEM (means) or Q1-Q3 (medians) and P-values in Tables.

We used analysis the Statistical Package for the Social Sciences. Data were added in tables.

  1. Several parts are repeated. Please delete them (L167-172, 275-276, 282-283, 428-430 etc)

We considered your suggestions.

  1. L195-198: This part refers to Methods. Please delete or shorten it.

We have considered your suggestions and moved to Methods. We shortened this text.

  1. L257-263: This section should be removed in Materials and Methods in L157 (last paragraph)

We moved text and added new subtitle “2.3. Last race”.

  1. L265-294: Please remove to the introduction (no results are presented)

The first four paragraphs from discussion were deleted.

  1. L431-458: This part is not related with your results and is partly mentioned in introduction. Please shorten it and try to connect it with your results

We considered your suggestions.

  1. Authors often state that “Pigeons that are under stress are more susceptible to infections”. Did you reach to this conclusion in your study? Where are the data that confirm this conclusion?

We added text: “Based on our limited results we can only speculate that there is no direct correlation between PiCV infection and CORT levels”.

Specific Comments:

L25, 69: “feed” instead of “food” changed

L27-28: “…welfare status. Animals should be in…” changed

L29: “such as” instead of “during” changed

L33: “…pigeons for the improvement of their condition and the preservation of their welfare…” changed

L36: “recorded” instead of “followed” changed

L57: Please rephrase “spheres of the animal world”? rephrased

L63-65: “Warzecha [4] indicated that these problems can affect many animals and government veterinarians should be actively involved in these activities.” changed

L67: “…(CORT) is the main…” changed

L67-68: “…immune and stress responses.” changed

L70-71: Please rephrase “and is associated with further support for adaptation of the stress response” rephrased

L72: “…the maximum level of CORT naturally occurs at daybreak…” changed

L75-76: “…especially if it is not strictly regulated [11,12].” changed

L87: “Zigo  [2] found…” changed

L99: “due to” instead of “given” changed

L109: “…improve conditions during racing flights and pigeon…” changed

L114: “per breeder” instead of “from each” changed

L121: Sexually mature? added in the text, pigeons were 3 to seven years old

L123-126: “Twenty-four pigeons from G4 group (six from each flock) were tested for CORT levels twice: within three to five hours after returning from the last race (G4a) and 30 days after the last race in the season (G4b). Pigeons from all categories were also tested for CORT concentration in both samplings.” changed

L128: Please delete “for” changed

L139: Please delete “Each of ninety-six pigeons was included in our study.” deleted

L141: “…collected for PiCV, APMV, AIV and CP determination. Blood…” changed

L156: “provided the” instead of “were given” changed

L227-230: “Within five hours after the race was finished, the average CORT level in G4a was 59.4 nmol/L, almost two-fold higher compared to those in the other three categories (G1, G2 and G3).” changed

Table 3: Please provide subscripts “a” or “b” after “G4” provided

L256: This Table refers to CORT levels? What about the other parameters?
We are of the opinion that we do not have enough data for a correct statistical analysis.

L265: “firstly” instead of “first” changed

L273-275: Please remove “[31]” after “Kerpal et al.” and delete “(2019)” changed

L278: “… have also been advanced [1].” changed

L290: “All these stressors…” changed

L294: “examined parameters” instead of “focus” changed

L298: “…health status and…” changed

L317: “collected” instead of “taken” changed

L328-331: Please rephrase rephrased

L343: “Due to the fact that” instead of “Because” changed

L369: Please delete deleted

L375: “…welfare status of racing…” changed

L380: “…was 0.34 μg/dl (9.81 nmol/L) and the maximum level was…” changed

L383, 386: “trained” instead of “training” changed

L388: “Exhaustion” instead of “Extraction” changed

L396: “recorded” instead of “taken” changed

L423: “…protect breeders’ investment.” changed

L432: “The fast and appropriate transport to the race is very important." changed

Comment: Table 2 was changed. The contents of columns 3, 4 and 5 are grouped in column 3. Some data were added in tables.

Thank you again for your suggestions.

Sincerely yours author and coauthors.

Alenka Dovč

Round 2

Reviewer 2 Report

The article is substantially improved. Authors made all the recommended amendments and I suggest the acceptance of the article.

Please make the following corrections:

L124: "...were fed with diets produced..."

L142-143: Please rephrase

L387-388: Please rephrase